# TinyMem: Condensing Multimodal Memory for Long-form Video Action Detection

## Abstract

Despite the great advances in video understanding with deep neural networks, current solutions still struggle with input videos that last for minutes, if not hours. To mitigate this issue, existing approaches typically build a memory cache with dense visual embedding on video transformers to model the long-range spatiotemporal dependencies. However, even with hundreds of extended memory tokens, their results remain unsatisfactory. In this paper, we argue that more compact yet informative memory embeddings can effectively improve performance. To this end, we introduce TinyMem, a model built upon tiny multimodal memory for long-form video action detection. In particular, we condense redundant video content into succinct descriptions to derive abstract text semantics. Subsequently, we integrate visual embedding condensed by regions with text embedding. TinyMem beats a range of state-of-the-art models on AVA v2.2, Epic-Kitchens-100 and Breakfast with highly condensed memory, *e.g.*, 37.4 mAP with TinyMem-24-12 on AVA v2.2 while using 5 times fewer memory tokens than the baseline with dense visual memory embedding.

## 1 Introduction

While analyzing contents in short-term videos with a duration of a few seconds has seen rapid progress in recent years Bertasius et al. (2021); Fan et al. (2021); Feichtenhofer et al. (2022); Tong et al. (2022), techniques still fall short of meeting the demands of real-world settings. For instance, in streaming services, a plethora of media files would last for minutes and even hours. It is challenging yet crucial to explore extending current techniques for short-term videos to ones with much longer duration Grauman et al. (2022); Wu & Krahenbuhl (2021); Yang et al. (2023). As a result, researchers introduce approaches in a similar spirit to the human visual system, where long-range visual information is not processed all at once but received and parsed sequentially in short segments over time. This motivates the task of online action detection Damen et al. (2022); Wang et al. (2021); Xu et al. (2021), which entails segmenting long videos into multiple short-term clips and then predicting corresponding labels within each segment arranged in temporal order.

As activities in long videos are strongly correlated yet located in different space-time locations, modeling each clip independently leaves enriched relationships over time unexploited. To address this issue, incorporating a memory cache into networks becomes a mainstream strategy for online action detection. In particular, our design is encouraged by approaches Ryoo et al. (2023); Song et al. (2023); Wu et al. (2022) that augment representations of video transformers with historical spatial-temporal clues from cached visual tokens. However, prior studies simply leverage the dense output video tokens of certain transformer layers as memory (*i.e.*, vanilla visual memory) . For long-form videos, however, we argue that this design inherently incorporates redundant and inefficient visual signals. In particular, when memory length scales up, vanilla visual memory results in a rapid growth of the number of tokens involved in attention computation and hence poses great challenges for video transformers in identifying the most helpful clue from historical information.

To address these issues, we draw inspiration from the human memory system, where visual stimuli would be converted into a compressed and low-resolution format in working memory Kwak & Curtis (2022). To explore the best practice of compact and informative memory embedding, innovatively, we employ textual descriptions to interpret the abstract semantics emerging from videos and project captions into tiny text token as semantic memory. Imitating the human memory system, we

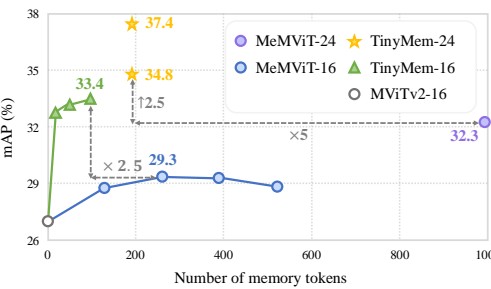
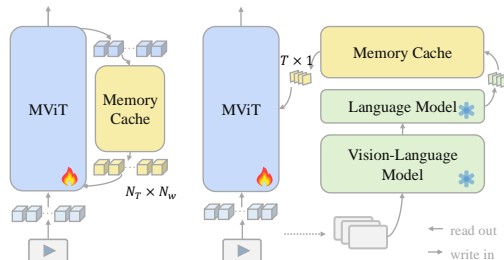

Figure 1: The trade-off between the average number of memory tokens used and mAP on AVA v2.2 when scaling up the memory length.

Figure 2: The comparison between the text memory cache of TinyMem and the vanilla visual memory cache of MeMViT.

supplement highly-abstracted text memory with imagery-connected memory Tulving et al. (1972). We hypothesize that employing tokens summarizing important object regions is sufficient to represent historical visual content as opposed to using all visual tokens in prior works. By incorporating the condensed multimodal memory, we introduce the **T**iny-**M**emory Multiscale Vision Transformer (TinyMem) for online action detection on long videos.

Specifically, we build our framework upon the short-term video transformer, MViTv2 Li et al. (2022). We exploit the powerful vision-language model (VLM), BLIP-2 Li et al. (2023) to generate framewise captions for the input clip. Subsequently, we project each obtained caption into **one tiny token** and write the resulting text token[1] of the entire clip into the memory cache, as illustarted in Fig. 2. Impressively, the single text memory token can effectively summarize the visual input and significantly reduce its dimensionality. In addition, we propose a captioner-free dynamic strategy to encourage the best efficiency during inference. As for region embedding, given off-the-shelf bounding boxes, we leverage ROI features as memory. Otherwise, we employ an extended set of global tokens as region tokens, and push the output region tokens into the memory cache. In implementation, we maintain a tiny number of memory tokens (*i.e.*, 16 tokens) to represent each video clip for each case. Therefore, with collaboration of these tow memory modalities, we manage to read out long duration of clip history from memory cahces effectively and efficiently. In consequence, TinyMem efficiently condenses video signals into a compact embedding space and achieves significantly improved performance thanks to streamlined memory tokens.

More importantly, we demonstrate the effectiveness of the proposed framework with extensive experiments. In particular, TinyMem-16-12 achieves **33.4** in mAP on AVA v2.2 Gu et al. (2018), which fuels MViTv2-16 by **6.4** and surpasses a range of prior methods with much larger vision backbones. As shown in Fig. 1, TinyMem-16-12 outweighs MeMViT-16 using vanilla visual embedding by **4.1** in mAP while using **2.5 times fewer** memory tokens. Besides, TinyMem-24-12 hits **37.4** in mAP, which outweighs MeMViT-24 by **5.1** and employs **5 times fewer** memory tokens. By using the captioner-free strategy in inference, TinyMem-24-12 generalizes well by achieving **34.8** without further fine-tuning, still striking advantage of **2.5** in mAP over MeMViT-24. We further validate the capacity of our method with additional action classification benchmarks, *e.g.*, Epic-Kitchens-100 Damen et al. (2022) and Breakfast Kuehne et al. (2014). Generally, we observe a consistent and promising growth in performance compared with the baselines, sufficiently proving the advantage of the proposed tiny memory design.

## 2 RELATED WORKS

**Online action detection** processes input videos as multiple streaming short clips arranged in temporal order An et al. (2023); Cao et al. (2023); De Geest et al. (2016); Wang et al. (2021; 2023a); Xu et al. (2021). The online framework especially benefits long-term video benchmarks De Geest et al. (2016); Idrees et al. (2017), *e.g.*, AVA Li et al. (2020), a dataset for spatial-temporal localization of atomic action, consisting of 15-minute-length videos. For instance, Zhao *et al.* replace vanilla attention with temporal smoothing kernels in Transformer Zhao & Krähenbühl (2022) to support

---

[1]Note that we use embedding, token, and memory interchangeably.

the arbitrary length of inputs. In this paper, we build our framework upon the online setting and aim to augment short-term video transformers with multimodal memory caches.

**Memory-augmented video models** cache motion information of historical video clips and are hence capable of modeling long-range spatial-temporal clues. Generally, prior strategies can be divided into two categories. The first vein of solutions builds memory networks upon a task-specific action decoder Chen et al. (2022); Huang et al. (2020); Tang et al. (2020); Wang et al. (2023a) that aggregates pre-extracted video features. LSTR Xu et al. (2021) compresses long-term video features and then cooperates the encoded memory with short-term video information. On the other hand, encouraged by memory techniques in NLP domain Dai et al. (2019); Rae et al. (2019); Wang et al. (2023c), researchers have been exploring strategies for plugging memory caches into video backbones Ryoo et al. (2023); Wu et al. (2022), especially on Transformers Vaswani et al. (2017). MeMViT Wu et al. (2022) caches the key and value tokens within attention layers of MViTv2 Li et al. (2022) and efficiently compresses memory tokens in a pipelined form. Nevertheless, distinct from previous methods, we introduce a novel modality, language, as the memory embedding, successfully incorporating highly-compact semantics with visual clues.

**Long-term video understanding** focuses on videos lasting for minutes and even hours Bahrami et al. (2023); Damen et al. (2018); Yeung et al. (2018); Tan et al. (2023); Yang et al. (2023). Recently, versatile explorations have been made to design specialized architectures Afham et al. (2023); Strafforello et al. (2023); Yu et al. (2020); Zhou et al. (2021; 2023) for processing long video sequences, *e.g.*, ViS4mer Islam & Bertasius (2022) takes advantage of a structured state-space sequence (S4) Gu et al. (2021) to efficiently aggregate long-range features of movie segments. Selective-S4 Wang et al. (2023b) further reduces the computational costs via adopting a selection network to drop tokens with less information before feeding into S4.

Additionally, researchers have been making attempts to leverage multimodal models for improved performance on long-form videos Argaw et al. (2023); Chen et al. (2023); Zhang et al. (2023b); Zhu et al. (2023b); Papalampidi et al. (2023). LF-VILA Sun et al. (2022) proposes to align video clips and text descriptions both temporally and globally via contrastive learning. While studies Yuan et al. (2023) have shown that advanced Vision-Language models (VLMs) Radford et al. (2021); Yu et al. (2022) lag behind state-of-the-art in action detection, our method manages to boost the performance by leveraging enriched language representation without extra video-language pre-training. TinyMem is also distinct from works Kim et al. (2023); Ren et al. (2024); Zhang et al. (2023a) exploiting the capacity of large language models for long-sequence understanding, as we focus on learning better visual representation of long-video and harness text embedding as the escalator.

## 3 METHOD

Our goal is to leverage multimodal memory to support online video action detection for long-form videos. To this end, we build our method, TinyMem, upon a short-term video model MViTv2 Li et al. (2022). Following the classical setting of online video processing, Given a streaming long video $\mathcal{V}$, we process it in a clip-by-clip manner. Specifically, at time step $t$, we could only get access to the current and past clips, *i.e.*, $\{\mathbf{C}_0, \cdots, \mathbf{C}_t\}$ that are arranged in a temporal sequence, and aim to predict the actions in $\mathbf{C}_t$. In the following, we begin with a brief review of the MViTv2 in Sec. 3.1, and then introduce the design of multimodal memory embedding in Sec. 3.2. Finally, we discuss memory reading and update strategy in Sec. 3.3 and introduce the captioner-free inference strategy in Sec. 3.4. The overall framework of our method is illustrated in Fig. 3.

### 3.1 A BRIEF REVIEW OF MViTv2

MViT Fan et al. (2021) brings the hierarchical design of convolution neural networks He et al. (2016) to vision transformers, via gradually downsampling the spatial resolution of visual features in stacked transformer blocks. On top of that, MViTv2 Li et al. (2022) further incorporates several advanced modules for better performance on a wide spectrum of visual tasks. Specifically, taking a video clip as input, MViTv2 first projects the 3D patches into a sequence of visual tokens $X$ via a patch embedding layer. After that, several transformer blocks are adopted to model the spatiotemporal relations between different tokens. Within each transformer block, $X$ are converted into query, key, and value through a linear projection $W_q$, $W_k$ and $W_v$, and then input to 3D pooling operations

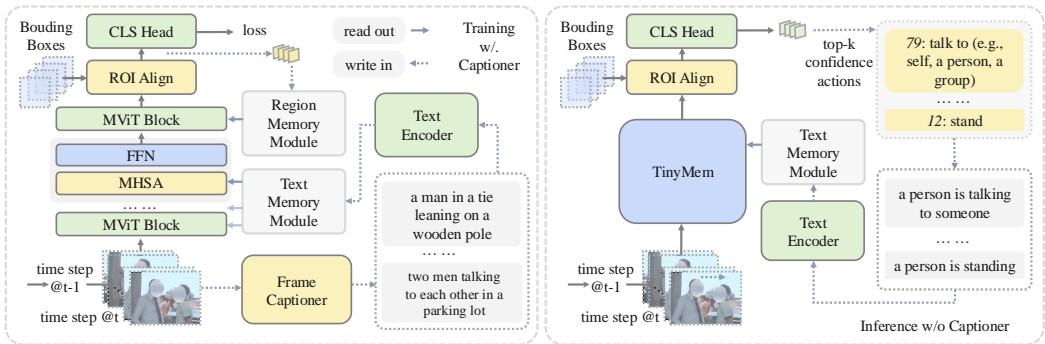

Figure 3: The overview of the TinyMem framework. **Left**: the training pipeline with BLIP-2 as the frame captioenr. **Right**: the captioner-free dynamic inference pipeline with region memory omitted for simplicity.

$\text{Op}_q$, $\text{Op}_k$ and $\text{Op}_v$. The pooled query, key, and value tokens $Q$, $K$ and $V$ then undergo multi-head self-attention computation.

Note that the pooling operation could effectively reduce the input sequence length and save the computation overhead in self-attention. Despite the impressive performance on clip-based video tasks, *e.g.*, video action recognition, MViTv2 cannot handle long videos with the absence of long-range context modeling. By cacheing the information of historical video clips in external multimodal memory modules, we effectively support long-from video modeling. In implementation, we follow Wu et al. (2022) to switch the sequence between pooling and linear projection.

## 3.2 MULTIMODAL MEMORY EMBEDDING

Existing memory-based video models, typically store dense visual features of past clips into memory, which contains remarkable redundant information and hence inevitably introduces distractors during memory reading. In this paper, we seek to comprise memory caches with abstract semantics by utilizing two modalities of memory embedding in TinyMem, *i.e.*, text and region memory.

**Text memory** stores the core concepts, *e.g.*, objects and their activities, in narrative captions. Specifically, given the current clip $C_t$, we obtain its description as a composition of captions from $T$ frames with BLIP-2 Li et al. (2023). Though BLIP-2 undergoes image-text pre-training only, owing to its outstanding zero-shot performance, it has been widely used to generate captions in recent works Bhattacharya et al. (2023); Chen et al. (2024); Yu et al. (2024) to assist video tasks. Subsequently, we embed generated captions into textual embedding using the text encoder of CLIP-B Radford et al. (2021). Next, we aggressively select a single [EOT] token to represent the full text semantics, and feed it into the projection head of vision-language joint feature space to obtain one memory tokens for each frame of $C_t$. The extracted $T$ text tokens condense video content into a highly compact space yet effectively model the context of past video segments.

**Region memory** caches the historical visual clues of informative regions. Compared with previous methods of storing pixel features of the complete clip, our region memory is more informative and efficient. Specifically, given bounding boxes predicted by an off-the-shelf object detector, RoI features are extracted by applying 3D RoI Align He et al. (2017) on the output visual tokens. We employ RoI features as compact region representation and feed them into the classification head to gain the final prediction. In consequence, we obtain resulting $N_{\text{region}}$ region memory tokens.

We argue that region memory can also be represented with an extended set of learnable tokens. Specifically, we prepend $N_{\text{region}}$ global tokens with input video tokens before the first transformer block and obtain the output global tokens on the corresponding layer as region embedding. We comprise region memory embedding with RoI features in default for spatiotemporal action detection benchmark and leverage global region tokens for temporal action detection benchmark.

**Multimodal memory** refers to the collaboration of text embedding and region embedding. Concretely, We select a subset of transformer layers and augment each layer with either of the above two types of memory embedding. Thereby, we inject the context information into the visual features of the current clip. Next, we will explain how to perform memory readout and update.

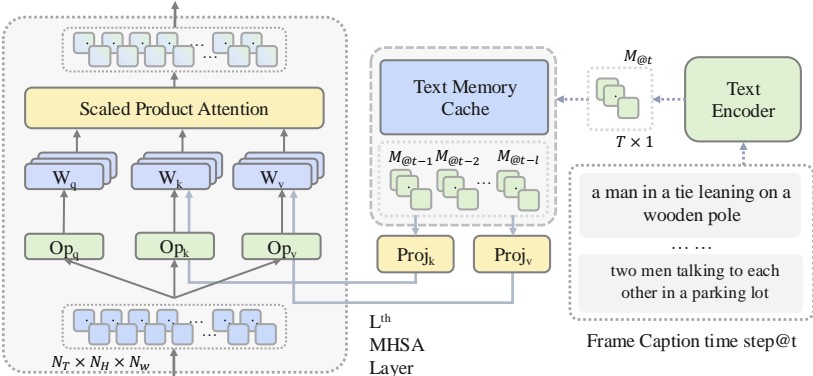

Figure 4: The readout and write-in pipeline of text memory cache on the L-th MHSA layer.

## 3.3 READ OUT & WRITE IN

In this section, we discuss the write-in and readout strategy of multimodal memory tokens. The proposed memory caches are inserted into a subset of transformer layers. Without loss of generality, we introduce the configuration for $L$-th layer and drop the layer index $L$ for simplicity, and visualize how text memory tokens interact with video tokens of the current time step in Fig. 4.

**Memory readout** supplements the visual features of the current clip with the information from previous clips. Specifically, we first sample $l$ memory tokens with a temporal stride of $s$ from the memory. To convert memory into "keys" and "values", we apply two lightweight projection layers $\mathcal{P}_k$ and $\mathcal{P}_v$ to align the feature dimension with the key and value tokens of the current clip, *i.e.*, $K_t$ and $V_t$. After that, we concat the obtained tokens of different timestamps along the sequence dimension and obtain $\mathcal{M}_k$, $\mathcal{M}_v$ respectively.

We implement $\mathcal{P}_k$ and $\mathcal{P}_v$ with a bias-free 1D group-convolutional layer followed by a shared normalization layer. It is worth noting that projection layers also compress the output length of our text (region) key and value tokens using a moderate stride size. More importantly, TinyMem obtains memory tokens much fewer than the visual memory ones in previous methods Wu et al. (2022). Next, we concatenate memory keys $\mathcal{M}_k$ and values $\mathcal{M}_v$ with $K_t$ and $V_t$ to get the memory-augmented key $K'$ and value $V'$ for self-attention computation.

**Memory write-in** is performed once the text embedding or region embedding is obtained by simply pushing them into the corresponding memory cache. We maintain both memory caches as First-In-First-Out (FIFO) queues with the maximal length equal to $L_c$. In other words, once the memory length exceeds $L_c$, the earliest cached memory tokens will be discarded.

## 3.4 CAPTIONER-FREE DYNAMIC INFERENCE

While we can access informative and high-quality captions with a wide range of powerful multi-modal foundation models now Bai et al. (2023); Liu et al. (2024); Zhu et al. (2023a), their model scale, *e.g.*, 7.96B parameters for BLIP-2 with OPT-6.7b Zhang et al. (2022), may introduce concerns for training and inference efficiency in resource-limit scenarios. To this end, we pre-extract frame captions for training videos and design a captioner-free strategy for inference. Specifically, we dynamically generate simplified captions according to the output predictions.

For video clip $C_t$, TinyMem outputs $N_{\text{box}}$ action predictions. We compute the product of the prediction confidence of each bounding box from the detector with the corresponding action prediction score. Next, we average the obtained product for each action and obtain Top-$K$ action categories $\{\text{act}_0, \cdots, \text{act}_K\}$. Afterward, we select the curated short description corresponding to each category in the format of "a person {performing some actions}". Later, we construct the text input for memory caches by randomly drawing from the $K$ obtained description. We claim that the simplified captions fall short of captions generated by BLIP-2 due to the absence of details. While we observe that TinyMem-24 still exhibits strong superiority and generalizes well on the captioner-free strategy.

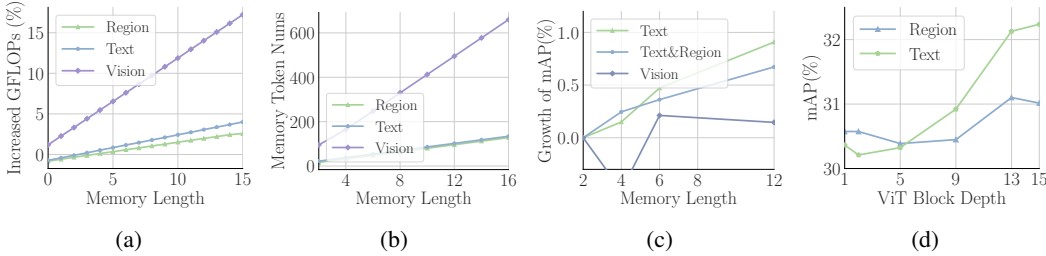

|  (a)  |  (b)  |  (c)  |  (d)  |

Figure 5: Comparison between scaling up different embedding. We present the influence of scaling up memory length over training GLOPs in Fig. 5a, the averaged number of memory tokens on average after the projection layer in Fig. 5b and mAP in Fig. 5c. Fig. 5d displays the impact of the depth of the selected layer when TinyMem-16-6 is augmented with only a single memory cache.

Table 1: Comparison between different memory embedding on TinyMem-16-6.

| Embedding | $N_{\text{mem}}$ | mAP |
|---|---|---|
| vison | $N_T \cdot N_H \cdot N_W$ | 30.0 |
| region (RoI) | $N_{\text{region}}$ | 31.0 |
| region (token) | $N_{\text{region}}$ | 31.7 |
| text (CLIP) | $T$ | 32.9 |
| vision (CLIP) | $T$ | 31.6 |

Table 2: Comparison between TinyMem and MeMViT on AVA v2.2.

| Model | mAP | $\bar{N}_{\text{mem}}$ |
|---|---|---|
| MeMViT-16-2 | 29.3 | 257 |
| TinyMem-16-2 | 32.8 (↑3.5) | 18 |
| MeMViT-24-2 | 32.3 | 992 |
| TinyMem-24-6 | 35.9 (↑3.6) | 96 |

## 4 EXPERIMENTS

**Model setup.** Notably, we explore two scales of architecture, denoted as TinyMem-$X$-$l$-$M$, with the suffix $X$ identifying the depth of the model, $l$ defining the memory length and $M \in \{V, R, T\}$ indicating the modality of memory embedding. In default, we omit "-$M$" and use the combination of text memory and region memory for TinyMem and adopt visual memory embedding for MeMViT. Besides, following MeMViT, we adopt the relative positional embedding and employ an attention-based head on MViTv2. Generally, when no memory is cached, TinyMem-$X$ are isotropic to MeMViT-$X$ in architecture.

For region memory embedding, we set $N_{\text{region}}$ to 16 and hence write in 16 text memory tokens for each clip. We adopt the ground-truth bounding boxes for training and detected person boxes from AIA Tang et al. (2020) for inference. To obtain clip descriptions, we adopt BLIP-2 models based on OPT-6.7b Zhang et al. (2022) and fine-tuned on COCO Lin et al. (2014). We apply the default prompt of "a photo of" to generate captions for each of $T$ frames. We observe that applying memory to deeper layers notably brings benefits, especially for region tokens (as shown in Fig. 5d). Therefore, we augment 50% of MHSA layers by employing region memory on the last sampled one and use text memory caches for all the rest.

**Implementation details.** We follow prior studies Wu et al. (2022); Xu et al. (2021) to process long videos by taking in a short clip containing one annotated keyframe each time step on AVA v2.2 . We fine-tune models[2] pre-trained on Kinetics Carreira et al. (2018); Kay et al. (2017) for 30 epochs by using an AdamW optimizer with a batch size of 128, a base learning rate of $5e^{-4}$ and a weight decay of 0.05. We resize the short size of frames to 256 and randomly crop a region of $224^2$ during training and apply random horizontal flipping and color jittering.

### 4.1 MAIN RESULTS

**Scaling up memory length** of TinyMem leads to performance gains effectively while maintaining a small number of memory tokens. Specifically, we compute the increase in GLOPs within the attention layers with memory length increased. Fig. 5a displays that the visual memory embedding consistently leads to more FLOPs while multimodal memory embedding maintains a growth of less than 5%. Similarly, we plot the growth of averaged memory token numbers in Fig. 5b and

---

[2]We adopt pre-trained checkpoints provided by MeMViT Wu et al. (2022).

Table 3: Comparison with SOTA methods on AVA v2.2.

| Model | Pretrain Dataset | Trainable Param | mAP |
|---|---|---|---|
| MViTv2-16, $16 \times 4$ Li et al. (2022) | K400 | 35M | 27.0 |
| MeMViT-16, $16 \times 4$ Wu et al. (2022) | K400 | 35M | 29.3 |
| AIA-R50, $4 \times 16$ Tang et al. (2020) | K700 | - | 29.8 |
| **TinyMem-16-12,** $16 \times 4$ | K400 | 42M | **33.4** |
| AIA-R101, $8 \times 8$ Tang et al. (2020) | K700 | - | 32.3 |
| STMixer-CSN152, $32 \times 3$ Wu et al. (2023) | K600 | - | 32.8 |
| MViTv2-24, $32 \times 3$ Li et al. (2022) | K600 | 51M | 30.1 |
| MeMViT-24, $32 \times 3$ Wu et al. (2022) | K600 | 53M | 32.3 |
| TubeR-CSN152, $32 \times 3$ Zhao et al. (2022) | K400† | - | 33.4 |
| STAR-ViT-B, $32 \times 3$ Gritsenko et al. (2023) | K700 | - | 33.9 |
| **TinyMem-24-12,** $32 \times 3$ | K600 | 58M | **37.4** |

Table 4: Ablation study of memory length and sampling stride in TinyMem-16.

| Memory Length ($l$) | Sample Stride ($s$) | mAP |
|---|---|---|
| $\times 2$ | $\times 2$ | 32.7 |
| $\times 4$ | $\times 2$ | 33.0 |
| $\times 6$ | $\times 2$ | 33.1 |
| $\times 12$ | $\times 2$ | 33.4 |
| $\times 12$ | $\times 1$ | 33.0 |
| $\times 12$ | $\times 2$ | 33.4 |
| $\times 12$ | $\times 3$ | 33.5 |
| $\times 12$ | $\times 4$ | 33.4 |

Table 5: Comparison with SOTA methods with comparable performance yet different model scales on AVA v2.2. † indicates using dynamic inference strategy without BLIP-2 generated caption.

| Model | Pretrain Dataset | Train Res. | Test Res. | Trainable Params | Full Params | mAP | Vid/s |
|---|---|---|---|---|---|---|---|
| **TinyMem-24-12,** $32 \times 3$ | K600 | 224 | 256 | 58M | 122M + 7.96B | **37.4** | 1.0 |
| **TinyMem-24-12†,** $32 \times 3$ | K600 | 224 | 256 | 58M | 122M | **34.8** | 5.9 |
| MeMViT-24, $32 \times 3$ | K600 | 224 | 256 | 53M | 53M | 32.3 | 5.5 |
| MViTv2-L, $40 \times 3$ | K700 | 312 | 312 | 213M | 213M | 34.4 | 4.9 |

the changes in mAP with memory length extended in Fig. 5c. The results showcase that multimodal memory tokens boil down informative clues within clips into a more compact feature space. Besides, extending text memory tokens effectively facilitates growth in performance, *e.g.*, mAP increases by 1.0 with $l$ increasing from 2 to 12. Similarly, Tab. 4 displays the scaling ability of TinyMem with mutimodal memory tokens.

**Text embedding overwhelms** other formats of embedding on AVA by a large margin, as indicated by Tab. 1. In specific, we compare the performance of different memory embedding with a uniform memory length of 6 and sampling stride of 1. $N_{\text{mem}}$ denotes the number of memory tokens read out before compression in each time step, with the layer index omitted. Among reported results, we encompass a special form of visual embedding that leverages the visual encoder of CLIP to generate compact visual embedding for each input frame. We project output visual tokens of ViT into visual-language joint space with the pre-trained head of the visual encoder in CLIP. Consequently, we obtain $T \times 1$ visual memory tokens for each input video clip.

Notably, RoI-based region embedding and CLIP visual embedding outperform the vanilla one by 1.0 and 1.6 in mAP, respectively, verifying that redundant information in plain visual memory can be substantially compressed. We also prove that wielding a set of global tokens as region embedding can be sufficiently effective, denoted as "region (token) " in Tab. 1. By default, we implement region embedding with extracted RoI features to avoid additional costs on AVA v2.2. More interestingly, text embedding achieves far superior performance, surpassing visual embedding by 2.9 in mAP and also exceeding CLIP visual embedding drawn from the aligned encoding space by 1.2. The absolute advantage demonstrates that the text modality can *effectively represent the semantics of spatiotemporal signals and has a remarkable capacity for compression*.

Furthermore, applying region memory on the deep layer of MViT achieves sufficiently good results, *e.g.*, TinyMem-16-6 achieves 31.1 mAP by augmenting the 13th layer with region memory, as displayed in Fig. 5d. This is similar to the result when augmenting 50% of the ViT layers in Tab. 1. Therefore, we combine text embedding with region embedding by plugging region memory into the late block of MViTv2. We observe that adding region embedding to pure text memory provides additional gains, as indicated by the steady growth of mAP in Fig. 5c. Specifically, compared with using text embedding alone, for each memory length, the combination of region embedding and memory embedding induces a growth of around 0.3 in mAP.

Table 6: Ablation study of text encoder and the text embedding design on TinyMem-16-6-T.

| Text Encoder | $N_{\text{mem}}$ $\times D_{\text{text}}$ | Proj. Head | mAP |
|---|---|---|---|
| CLIP-B | $T \times 76 \times 512$ | | 31.7 |
| | $T \times 32 \times 512$ | | 31.6 |
| | $T \times 1 \times 512$ | | 32.7 |
| | $T \times 1 \times 512$ | ✓ | 32.9 |
| CLIP-L | $T \times 1 \times 768$ | ✓ | 32.2 |
| T5-Base | $T \times 1 \times 768$ | | 31.9 |
| | $T \times 1 \times 768$ | ✓ | 31.9 |

Table 7: Ablation study of memory compression factor and compression kernel on TinyMem-16-12.

| Embedding | Compress Factor | Compress Kernel | mAP |
|---|---|---|---|
| text | $\times 1$ | $\times 1$ | 32.7 |
| | $\times 2$ | $\times 3$ | 33.0 |
| | $\times 4$ | $\times 7$ | 32.6 |
| region | $\times 1$ | $\times 1$ | 31.0 |
| | $\times 2$ | $\times 3$ | 31.3 |
| | $\times 4$ | $\times 7$ | 31.3 |

**State-of-the-art performance.** We compare our models with MeMViT, a top-performing model based on vanilla visual memory, as shown in Tab. 2, and report the average number of memory tokens involved in each self-attention layer, denoted as $\bar{N}_{\text{mem}}$. With the same memory length, TinyMem-16-2 outperforms its counterpart by **3.5** in mAP and consumes 14 times fewer extended tokens. On the other hand, TinyMem-24-6 adopts 3 times longer memory, yet maintains 10 times fewer memory tokens than MeMViT-24-2 and surpasses the latter by **3.6** in mAP.

Additionally, we display results of TinyMem and prior SOTA models on AVA v2.2 in Tab. 3. Notably, TinyMem-16 delivers the mAP of **33.4**, beating a range of models with much fewer trainable parameters, and a much smaller scale of pre-training datasets. Similarly, TinyMem-24 hits the mAP of **37.4**, ranking first in the table. Moreover, as shown in Tab. 5, while TinyMem-24 surpasses MViTv2-L with a significantly larger scale of trainable parameters. We attribute the leap in performance to the powerful multimodal memory tokens, especially the abstract text embedding.

**Captioner-free inference strategy**, as introduced in Sec. 3.4, helps model to remove the computational costs of BLIP-2 and effectively enhance the inference speed. While TinyMem-24 experiences a performance drop due to the distribution shift between generated captions dependent on action predictions and captions generated by BLIP-2 during training, the strategy still helps TinyMem-24 to beat MeMViT-24 by a large margin, *i.e.*, 2.5 in mAP and achieves higher throughput. We claim that the captioner-free inference strategy is highly correlated to the generalization capability of the model and discuss further details about the strategy in the Appendix.

## 4.2 ABLATION STUDIES

In this section, we investigate the influence of different designs in TinyMem. Hereby, we present ablation experiments behind the core design and highlight the default setting with gray in the table.

**Text encoder.** Tab. 6 displays the results of adopting different text encoders for TinyMem-16. For all experiments, we employ text embedding as memory only. We pad or truncate input captions to the length of $\mathcal{N}$ in tokenization when $\mathcal{N} > 1$. For CLIP, the projection head refers to the pre-trained linear projection that maps text features into the vision-language joint space. For T5, we initiate an additional learnable linear layer as the projection head for training.

We observe that equipping text embedding encoded by CLIP-B surpasses the feature of T5-Base by 0.9 in mAP. Generally, we contribute the edge of CLIP over T5 to its vision-language pre-training strategy. Besides, keeping only one text token output of CLIP-B yields the best results with reduced computational costs. We believe that text embedding possesses a promising compression capacity, therefore it effectively supports an aggressive reduction of tokens. Tab. 6 also demonstrates that leveraging CLIP-L as the text encoder underperforms the CLIP-B-based counterpart by 0.6 in mAP. TinyMem yields the best performance when accessing the text encoder with a similar scale and performs worse with a text encoder scaling up.

**Region embedding.** Tab. 9 displays the result of adopting a different number of region tokens. Generally, the number of ground-truth bounding boxes for each keyframe is less than 16 in AVA v2.2. Extending the number of region tokens to above 16 leads to a performance drop of 0.3 and an

Table 8: Comparison between TinyMem, MViTv2 Li et al. (2022) and MeMViT Wu et al. (2022) on action recognition benchmark of Breakfast.

| Model | mAP | Model | mAP |
|---|---|---|---|
| MViTv2-16 | 29.5 | MViTv2-24 | 34.1 |
| MeMViT-16-4 | 47.8 (↑18.3) | MeMViT-24-4 | 48.8 (↑14.7) |
| TinyMem-16-4 | 48.7 (↑19.2) | TinyMem-24-4 | 52.4 (↑18.3) |

Table 9: Ablation study of the number of region memory tokens on TinyMem-16-12-R.

| $N_{\text{region}}$ | mAP |
|---|---|
| 8 | 31.1 |
| 16 | 31.3 |
| 32 | 31.0 |

Table 10: Comparison between TinyMem, MViTv2 Li et al. (2022) and MeMViT Wu et al. (2022) on action classification benchmark of Epic-Kitchens-100 .

| Model | Verb | Noun | Action | Model | Verb | Noun | Action |
|---|---|---|---|---|---|---|---|
| MViTv2-16 | 70.0 | 56.0 | 45.1 | MViTv2-24 | 72.6 | 59.6 | 48.6 |
| MeMViT-16-4 | 70.7 (↑0.7) | 56.7 (↑0.7) | 45.7 (↑0.6) | MeMViT-24-4 | 72.8 (↑0.2) | 61.8 (↑2.2) | 50.5 (↑1.9) |
| TinyMem-16-4 | 71.3 (↑1.3) | 58.5 (↑2.5) | 47.5 (↑2.4) | TinyMem-24-4 | 73.3 (↑0.7) | 62.4 (↑2.8) | 51.2 (↑2.6) |

increase in computational costs. Nevertheless, decreasing the number to 8 also leads to a decrease in mAP. In default, we set $N_{\text{region}}$ to 16 for TinyMem.

**Projection module.** Tab. 7 presents the influence of different convolution designs for the memory projection module. We investigate text and region memory independently by adopting one format of embedding for all involved layers. The compression factor corresponds to the stride size of convolution and the compression kernel refers to the kernel size of convolution. By default, we choose the compression convolution with a kernel size of 3 and a stride size of 2.

### 4.3 GENERALIZATION ANALYSIS

**Results on Breakfast.** We further examine the proposed framework on action recognition benchmark of Breakfast Kuehne et al. (2014) dataset, where video samples of preparing breakfast last for 2.3 minutes on average and are annotated with 48 sub-action classes and non-overlapping timestamps for the start and the end of each action. We follow the setting of online action detection and predict the action label assigned for each annotated clip segment within long videos. Generally, we keep using the rolling attention mask strategy and employ 8 global tokens for region memory.

As shown in Tab. 8, multimodal memory embedding significantly improves the results of clip-wise action classification. Meanwhile, TinyMem-16-4 outperforms MeMViT-16-4 by 0.9 in mAP, indicating the advantage of multimodal memory. Notably, TinyMem-24-4 surpasses its counterpart by **3.6** in mAP, leading to a growth of **18.3** over the MViTv2-24 baseline.

We claim that our method can also be easily adapted for long-term activity detection, where models output all action categories within the complete video. We extract 256 frames for each video sample and divide it into clips containing 16 frames in the temporal order and input them sequentially into the model. For baseline without memory, we average the prediction of each clip and report the prediction result of the last clip for memory-augmented models. As Tab. 12 shows, both TinyMem and MeMViT surpass the MViTv2 baseline, while the proposed multimodal memory outperforms the vanilla visual embedding by **2.0** in mAP. Moreover, TinyMem-16 with compact region embedding surpasses the one using joint multimodal memory embedding. We contribute the results to that frozen CLIP embedding may results in overfitting on small-scale dataset.

**Results on Epic-Kitchens-100.** We explore the efficacy of multimodal memory embedding on Epic-Kitchens-100 Damen et al. (2018) action classification benchmarks. Instead of extracting additional bounding boxes and RoI features, we prepend 8 learnable global tokens as region embedding. Videos in Epic-Kitchens-100 vary in length and hence cause unaligned memory caches for video clips within the same mini-batch. To avoid involving inconsistent historical information by mistake, we apply a rolling attention mask for training, which masks out memory tokens from different videos and updates via sliding when memory tokens in caches reach the maximum length.

Table 11: Comparison between TinyMem-ViT and VideoMAE Tong et al. (2022)with different memory embedding on AVA v2.2.

| Model | Mem | mAP |
|-------|-----|-----|
| VideoMAE-S | - | 28.4 |
| | V | 28.9 (↑0.5) |
| | R | 29.8 (↑1.4) |
| TinyMem-ViT-S-12 | T | 31.7 (↑3.3) |
| | T + R | 32.2 (↑3.8) |
| VideoMAE-B | - | 31.8 |
| TinyMem-ViT-B-12 | T + R | 33.7 (↑1.9) |

Table 12: Comparison with SOTA methods on the long-term activity detection benchmark of Breakfast.

| Model | mAP |
|-------|-----|
| Timeception-RN50 Hussein et al. (2019a) | 59.6 |
| VideoGraph-I3D Hussein et al. (2019b) | 63.1 |
| GHRM-I3D Zhou et al. (2021) | 65.9 |
| MViTv2-16 Li et al. (2022) | 62.6 |
| MeMViT-16-4 Wu et al. (2022) | 63.8 |
| TinyMem-16-4 | 65.8 |
| TinyMem-16-4-R | **66.6** |

To ensure a fair comparison, we follow the implementation of MeMViT to perform training and report reproduced results of MeMViT as the baseline with vanilla visual memory. As demonstrated in Tab. 10, TinyMem-16-4 outperforms MeMViT-16-4 on Top-1 accuracy by 0.6%, 1.6% and 1.6% on verb, noun and action class respectively. Likewise, TinyMem-24-4 improves MeMViT-24-4 by 0.5%, 0.6% and 0.7% in accuracy.

**Multimodal memory on ViT.** We believe that our framework can be easily adapted for other ViT variants. Typically, we explore TinyMem-ViT upon VideoMAE Tong et al. (2022). We follow the fine-tuning setting of VideoMAE on AVA by assuming sinusoidal positional embedding on video tokens while keeping relative positional embedding between query tokens and memory tokens.

We train TinyMem-ViT in an identical training setting with TinyMem, and set memory length $l$ to 12 and memory step to 1 for reported results of small and base models. Besides, we keep to the policy of combining text embedding and region embedding by employing region memory on the last selected layer and augmenting the remaining 50% layers with text memory caches. Note that in contrast to TinyMem-24, TinyMem-ViT-24 tasks in 16 video frames with a sampling stride of 4.

Furthermore, we propose that our framework can be generalized for more transformer-based architecture. Hereby, we explore the condensed multimodal memory embedding on VideoMAE Tong et al. (2022) pre-trained models. As displayed in Tab. 11, applying vanilla visual embedding on ViT brings about marginal growth in mAP while using region embedding or text embedding leads to more solid gains. Additionally, employing the proposed multimodal memory receives an increase of **3.7** over the VideoMAE-S baseline. Similarly, TinyMem-ViT-B gives rise to an increase of **2.1** compared with VideoMAE-B. Besides, We fine-tune the visual encoder of CLIP-B/16 Radford et al. (2021) with the same training setting and achieved 12.5 in mAP. We observe that the direct adaptation fails to achieve a competitive result.

## 5 CONCLUSION

Supporting top-performing video models on long video becomes an essential research topic concerning the practical demands of long-form video understanding. While explorations have been made to augment short-term models with different memory designs, prior methods cache redundant visual information only. In this paper, we innovatively included text, *i.e.*, brief descriptions of video contents, as the memory modality for spatiotemporal signals. Surprisingly, the abstract format of embedding demonstrated strong representation capabilities with highly compact dimensions. On top of that, we proposed TinyMem, which jointly employed compact text embedding and largely concentrated visual embedding as memory, and achieved promising performance on long-term action detection benchmarks including AVA v2.2, Epic-Kitchens-100 and Breakfast.

Due to constraints of resources, we didn't explore more long-form video tasks (*e.g.*, temporal action localization, long video question answering and *etc*) and models with larger scales. We believe that our work can inspire more explorations in vision-language model, especially multimodal large language models, *e.g.*, combining frame description or compact region embedding with videos as input or memory to enhance performance of models. Nevertheless, we believe that TinyMem opens up new possibilities by leveraging multimodal embedding for more long-term video tasks.

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

## A  IMPLEMENTATION DETAILS

In the following, we discuss the relative position embedding strategy and training configurations on Epic-Kitchens-100 Damen et al. (2018) and Breakfast Kuehne et al. (2014) benchmarks.

**Implementation details.** By default, we report the performance of TinyMem-16 taking in clips with 16 frames sampled at a temporal stride of 4 and TinyMem-24 with 32 frames sampled at a temporal stride of 3 as input. During inference, we set the batch size to 1 to feed all test samples in the online mode and resize the short size of frames to 256. For scaling experiments, we compute FLOPs using `fvcore` on frames sized of $224^2$ on the same NVIDIA 80-GB A100 GPU.

**Relative position embedding.** We leverage relative position embedding on the attention map between current video tokens and extended memory keys. Following Wu *et al.* Wu et al. (2022), we employ relative positional embedding on current video tokens and memory tokens separately. Specifically, we operate the overall position embedding as $\mathbf{R} = \texttt{concat}(R', R)$, with $R$ standing for embedding between video visual tokens and $R'$ denoting the one between video tokens and memory tokens. The implementation can be formulated as

$$\text{Attn} = \text{Softmax}(QK'^T + \mathbf{R}/\sqrt{D})V',$$
$$R_{ij} = Q_i \cdot (R^t_{t^q_i, t^k_j} + R^h_{h^q_i, h^k_j} + R^w_{w^q_i, w^k_j}),$$
$$R'_{ij} = Q_i \cdot R_{t^q_i, t^{m_k}_j},$$

where $t^q_i$, $t^k_i$ and $t^{m_k}_j$ represent the temporal positions of token $i$ in current query tokens, token $j$ in current key tokens and token $j$ in memory "key" tokens. Besides, $h$ and $w$ denote the relative positions along the height and width of video tokens.

Table 13: Comparison with SOTA methods on EK-100.

| Model | Pretrain | Verb | Noun | Action |
|---|---|---|---|---|
| Omnivore Girdhar et al. (2022) | IN+K400+SUN | 69.5 | 61.7 | 49.9 |
| MTV Yan et al. (2022) | WTS-60M | 69.9 | **63.9** | 50.5 |
| MeMViT-24* Wu et al. (2022) | K600 | 72.8 | 61.8 | 50.5 |
| LaviLa-L Zhao et al. (2023) | WIT+Ego4D | 72.0 | 62.9 | 51.0 |
| TinyMem-24-4 | K600 | **73.3** | 62.4 | **51.2** |

**Epic-Kitchens-100.** We follow MeMViT Wu et al. (2022) to train our models on Epic-Kitchens-100 for 30 epochs with an AdamW Loshchilov & Hutter (2018) optimizer and adopt a weight decay of 0.05, a batch size of 128 while finding that the base learning rate of $5e^{-4}$ yields better results. Note that we initialize our models with weights of MViTv2 pre-trained on Kinetics-400 Kay et al. (2017) and Kinetics-600 Carreira et al. (2018).

Table 14: Ablation study of different multimodal embedding designs on TinyMem-16-12.

| Augmented Layer | Layerwise Embedding | mAP |
|---|---|---|
| $[0, 2, 4, 6, 8, 10, 12, 14]$ | $[R, T, R, T, R, T, R, T]$ | 32.8 |
| $[0, 1, 2, 4, 8, 13, 14, 15]$ | $[T, R, T, R, T, T, T, T]$ | 33.2 |
| $[0, 2, 8, 10, 12, 13, 14, 15]$ | $[T, T, T, T, T, T, T, R]$ | 33.4 |
| $[0, 2, 8, 10, 12, 13, 14, 15]$ | $[T, T, T, T, T, T, R, R]$ | 33.4 |
| $[0, 2, 8, 10, 12, 13, 14, 15]$ | $[T, T, T, T, T, R, R, R]$ | 32.5 |

Table 15: Ablation study of frame captioner on TinyMem-16-12-T.

| Frame Captioner | mAP |
|---|---|
| Tag2Text Huang et al. (2023) | 31.4 |
| BLIP2-opt6.7B-coco Li et al. (2023) | 33.0 |
| BLIP2-flant5-xl-coco Li et al. (2023) | 33.0 |
| w/o. | 30.0 |

We apply Rand Augment Cubuk et al. (2020) with a probability of 0.5 for 4 layers of maximum magnitude 7, label smoothing Szegedy et al. (2016) with a magnitude of 0.01 and random erasing Zhong et al. (2020) with a probability of 0.25. Similarly, we adopt two classification heads to generate predictions for verb and noun respectively. We also find that applying augmentation for text inputs benefits the performance. Specifically, we delete or swap 1-2 words in 30% randomly drawn captions with the implementation of nlpaug Ma (2019).

In experiments, we observe that the model trained with captions generated by BLIP-2 surpasses the performance of one trained with LaViLa captioner. Moreover, the results also demonstrate that compact region memory brings more benefits than text memory. We speculate that text semantics might be less accurate and discriminative for fine-grained actions on a narrow domain, *i.e.* activities in kitchens. In default, we infuse 75% of the selected layer with the condensed region embedding and inject text embedding into the other 25% of the selected layers. As displayed in Tab. 13, TinyMem-24 surpasses state-of-the-art methods on action category on the action classification benchmark without large-scale pre-training.

**Breakfast.** We train our models on Breakfast Kuehne et al. (2014) for 20 epochs with an AdamW Loshchilov & Hutter (2018) optimizer and adopt a weight decay of 0.05, a batch size of 64, and a learning rate of $2.5e^{-4}$. Generally, we adopt an identical configuration of augmentations with one of Epic-Kicthens-100 training. For training samples, we resize the short size of frames to 256 and randomly crop a region of $224^2$. During inference, we resize the short side size of test video clips to 256, and center crop each video frame to the resolution of $224^2$. Similarily, in default, we infuse 75% of the selected layer with the condensed region embedding and inject text embedding into the other 25% of the selected layers.

# B  MORE ABLATIONS

**Frame captioner.** Tab. 15 presents the influence of different frame captioning model. Specifically, we leverage two powerful vision-language models, tag2text Huang et al. (2023) and BLIP-2 Li et al. (2023) to generate framewise captions in a zero-shot setting. We cap the maximum length of output tokens for the text decoders in BLIP-2 to 48 and one for the text decoder in tag2text to 50. Nevertheless, the average output sentences for these two models vary in length, with length hereby referring to the number of characters in each description. While BLIP-2 generally generates shorter and coarser captions, it achieves better outcomes. We posit that shorter captions obtain more condensed semantics in encoding and preserve more informative tokens in compression.

**Combination of compact embedding.** Tab. 14 shows TinyMem-16-12 with different configuration of text embedding and region embedding. "T" denotes for text embedding and "R" stands for using region embedding on the corresponding layer. As shown in the experimental results, alternatively applying two forms of embedding performs worse than applying the region embedding in last layers. Nevertheless, applying region memory at the beginning stage hardly produces benefits. On the other hand, plugging in region embedding on the last ViT blocks effectively improves the performance.

**Captioner-free inference.** We convert action classes in AVA v2.2 to simple descriptions, as displayed in Tab. 16. As the strategy employs frame captions which miss out on detailed information and are shorter in length compared to BLIP-2-generated captions as shown in Fig. 8, the performance of the model is highly related to the generalization capability. Therefore, we observe that TinyMem-24 model experiences less performance drop (as displayed in Tab. 4) while TinyMem-16 decreases by 3.0 in mAP, as listed in the last row of Tab. 15.

Table 16: The action description of AVA v2.2 for captioner-free inference.

| ID | Description | ID | Description |
|---|---|---|---|
| 0 | 'a person bend or bow at the waist' | 1 | 'a person crawl' |
| 2 | 'a person crouch or kneel' | 3 | 'a person dance' |
| 4 | 'a person fall down' | 5 | 'a person get up' |
| 6 | 'a person jump or leap' | 7 | 'a person lie or sleep' |
| 8 | 'a person martial art' | 9 | 'a person run or jog' |
| 10 | 'a person sit' | 11 | 'a person stand' |
| 12 | 'a person swim' | 13 | 'a person walk' |
| 14 | 'a person answer phone' | 15 | 'a person brush teeth' |
| 16 | 'a person carry or hold an object' | 17 | 'a person catch an object' |
| 18 | 'a person chop' | 19 | 'a person climb a mountain' |
| 20 | 'a person clink glass' | 21 | 'a person close a door, a box' |
| 22 | 'a person cook' | 23 | 'a person cut' |
| 24 | 'a person dig' | 25 | 'a person dress or put on clothing' |
| 26 | 'a person drink' | 27 | 'a person drive a car, a truck' |
| 28 | 'a person eat' | 29 | 'a person enter' |
| 30 | 'a person exit' | 31 | 'a person extract' |
| 32 | 'a person fishing' | 33 | 'a person hit an object' |
| 34 | 'a person kick an object' | 35 | 'a person lift or pick up' |
| 36 | 'a person listen to music' | 37 | 'a person open a window, a car door' |
| 38 | 'a person paint' | 39 | 'a person play board game' |
| 40 | 'a person play musical instrument' | 41 | 'a person play with pets' |
| 42 | 'a person point to an object' | 43 | 'a person press' |
| 44 | 'a person pull an object' | 45 | 'a person push an object' |
| 46 | 'a person put down' | 47 | 'a person read' |
| 48 | 'a person ride a bike, a car, a horse' | 49 | 'a person row boat' |
| 50 | 'a person sail boat' | 51 | 'a person shoot' |
| 52 | 'a person shovel' | 53 | 'a person smoke' |
| 54 | 'a person stir' | 55 | 'a person take a photo' |
| 56 | 'a person text on or look at a cellphone' | 57 | 'a person throw' |
| 58 | 'a person touch an object' | 59 | 'a person turn a screwdriver' |
| 60 | 'a person watch TV' | 61 | 'a person work on a computer' |
| 62 | 'a person write' | 63 | 'a person fight or hit a person' |
| 64 | 'a person give or serve an object to a person' | 65 | 'a person grab a person' |
| 66 | 'a person hand clap' | 67 | 'a person hand shake' |
| 68 | 'a person hand wave' | 69 | 'a person hug a person' |
| 70 | 'a person kick a person' | 71 | 'a person kiss a person' |
| 72 | 'a person lift a person' | 73 | 'a person listen to a person' |
| 74 | 'a person play with kids' | 75 | 'a person push another person' |
| 76 | 'a person sing to self, a person, a group' | 77 | 'a person take an object from a person' |
| 78 | 'a person talk to self, a person, a group' | 79 | 'a person watch a person' |

## C   MORE ANALYSIS

**Visualizations.** In Fig. 8, we display captions generated by BLIP-2 for AVA v2.2 in the zero-shot setting. Though the quality of generated captions is generally coarse, noisy descriptions still serve as an effective memory agent. Additionally, as indicated by Fig. 8, the generated captions are coarse in quality since the generation is performed in the zero-shot setting. Still, the noisy descriptions serve as an effective memory agent. As displayed in Fig. 6, in the current clip, a pastor is giving a speech with a hand holding a cross. While generated captions mistake the cross as "a cigarette", "a

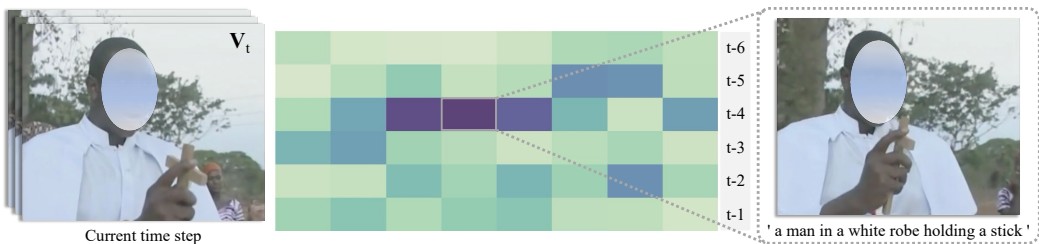

Figure 6: Visualization of attention map between the `CLS` token and text memory keys.

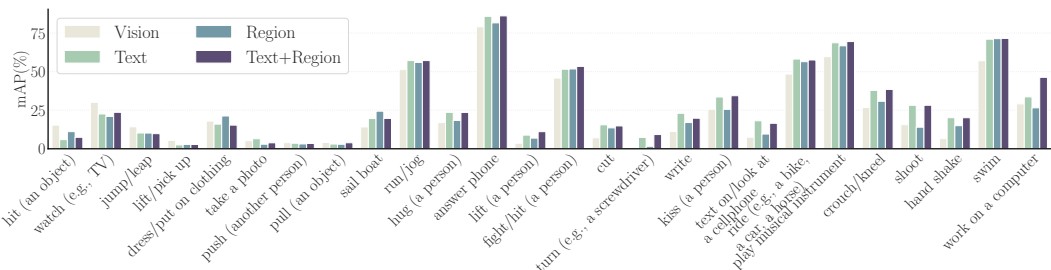

Figure 7: Comparison between different memory embedding on specific action class.

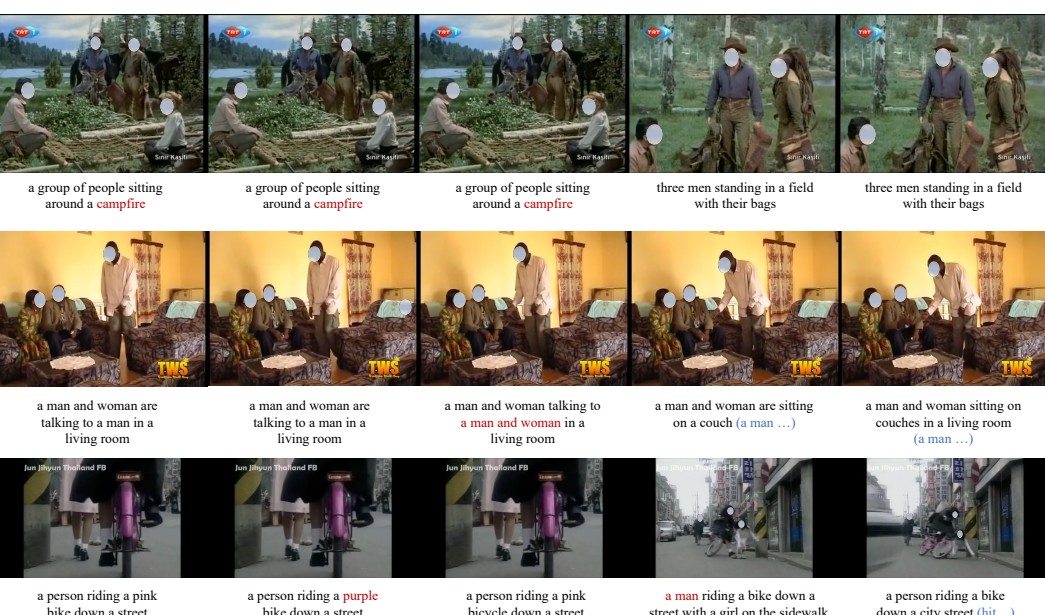

Figure 8: Visualization of description examples for AVA and their corresponding video frames. Generally, BLIP-2 generates short captions that describe major activities in each frame and contain noise and mistakes, as red color indicates incorrect words and blue color represents missing information.

microphone" or "a toothbrush." The caption with the highest attention score corresponds to "a man in a white robe holding a stick", conveying the closest semantics.

We compare the impacts of different memory embedding on each action category on AVA and plot the ones dropping and growing most in Fig. 7. Generally, multimodal embedding surpasses vanilla visual embedding in almost all categories except for "hit (an object) ", "push (another person)", "pull (an object)" and "watch (*e.g.*, TV)." We posit that frame-based description cannot distinguish between ambiguous actions, *e.g.*, push and pull, and may fail to capture big spatiotemporal move-

ments, *e.g.*, hit. We hypothesize that clip-based captions generated by video-language models would bring about greater improvements.

