# OpenReview forum: "TinyMem: Condensing Multimodal Memory for Long-form Video Action Detection"
_ICLR.cc/2025/Conference — ICLR 2025 Conference Withdrawn Submission_

### Official Review · Reviewer_qsbc · 2024-10-25

**Soundness:** 3
**Presentation:** 3
**Contribution:** 2
**Rating:** 5
**Confidence:** 4

**Summary:**

The paper **TinyMem: Condensing Multimodal Memory for Long-Form Video Action Detection** proposes TinyMem, a model that efficiently condenses video frames into a compact embedding for online action detection. It introduces a novel multimodal memory design that combines visual and text embeddings to reduce the memory footprint while maintaining or improving performance. The key contribution is using language descriptions (captions) and condensed visual regions as memory tokens, which significantly reduces the number of tokens required. The model is evaluated on long-form video benchmarks such as AVA v2.2, Epic-Kitchens-100, and Breakfast, showing superior performance compared to prior models like MeMViT.

**Strengths:**

**Novelty in Memory Design**: Introducing text-based memory tokens alongside visual tokens is a novel approach. By using compact textual representations, the model reduces memory usage while preserving critical semantic information. The results are intriguing as just using compressed textual information leads to such as an improvement on the evaluated tasks.

**Memory Efficiency**: TinyMem achieves competitive or superior performance with fewer memory tokens. The model maintains high accuracy on benchmarks such as AVA, Epic-Kitchens and Breakfast while reducing the computational overhead, which is crucial for long-form video understanding understanding.

**Weaknesses:**

Overall, the proposed method is simple, with limited novelty, yet the results are intriguing. The discussion here covers key aspects of the approach, along with potential limitations and questions that arise from the study’s findings.
Here is a list of key limitations/questions.

#### 1. Captioner Dependence
The model demonstrates efficiency, but its training is dependent on captions generated by models such as BLIP-2. This reliance could introduce external dependencies and may lead to significant computational costs during training. The implications of these dependencies are worth examining, especially concerning scalability and robustness.

#### 2. Captioner-Free Dynamic Inference
The process of Captioner-Free Dynamic Inference remains unclear, specifically how it avoids leaking label information. During inference, captions are generated heuristically using predicted action labels, which simplifies computation. However, this approach raises questions about robustness. Incorrectly predicted actions could lead to flawed captions, potentially compounding errors throughout the inference process.

#### 3. Temporal Relationships in Captioning
A notable flaw in the captioning method is its disregard for temporal relationships between frames, which might limit the model’s ability to capture nuanced temporal dynamics within videos.

#### 4. Text Memory Compression Technique
The technique used for text memory compression (Section 3.2) is highly aggressive, condensing an entire caption into a single token. The authors should clarify why they chose the `[EOT]` token for this purpose and discuss potential outcomes if tokens were sparsely sampled or averaged. Would these approaches lead to improved performance? Alternatively, why not consider taking the average of all token embeddings after mapping to a joint vision-language feature space?

#### 5. Token Merging Techniques in Table 2
In Table 2, would results differ if visual tokens were merged using techniques like average pooling or cosine similarity? A comparison of these methods might provide insights into token merging strategies and their impact on model accuracy.

#### 6. Effect of Additional Textual Data
The results in Table 1 indicate that additional textual data improves model performance over other information types, such as ROI visual tokens. Is this due to the auxiliary data source, or are there other contributing factors? The authors could provide an analysis of this observation.

#### 7. Scalability of Memory for Long Videos
Further discussion is needed on how the model’s memory mechanism scales with longer videos. This is particularly relevant in cases where extended temporal context might affect performance or computational feasibility.

#### 8. Choice of Comparison Model
The study uses VideoMAE as a comparison model in Table 11. However, clarification is needed on why VideoMAE was selected. Is this model chosen due to its distinct architecture, such as vanilla ViT or MViT, or because it features self-supervised pretraining? Additionally, was TinyMem initialized from a similarly pretrained model?

#### 9. Example Captions and Ground Truth Comparisons
It would be beneficial to examine some captions generated by BLIP-2 alongside corresponding video frames and the ground truth labels. Such comparisons could offer insights into caption accuracy and alignment with true actions in the video content.

#### 10. Limited Task Exploration
The evaluation is limited to long-form video benchmarks, though the study briefly mentions the possibility of exploring other tasks, like video question-answering and temporal action localization. Expanding the scope of tasks could provide a more comprehensive evaluation of the model’s versatility and robustness across diverse video-based applications.

### Missing Related Work on Memory

The approach closely resembles the Just Caption Every Frame (JCEF) baseline, yet a direct comparison is absent. Including this would help in evaluating the model’s novelty and performance against established baselines. Additionally, references to related work, such as [1] and [2], are missing.

---

### References

1. Min et al., *MoReVQA: Exploring Modular Reasoning Models for Video Question Answering*, CVPR 2024.
2. Kahatapitiya et al., *VicTR: Video-conditioned Text Representations for Activity Recognition*, CVPR 2024.

**Questions:**

See weakness.

---

### Official Review · Reviewer_aqMY · 2024-10-30

**Soundness:** 2
**Presentation:** 1
**Contribution:** 2
**Rating:** 3
**Confidence:** 4

**Summary:**

TinyMem addresses a significant challenge in video understanding: the ability to process and analyze lengthy videos that span minutes or hours. While current deep learning models excel at analyzing short video clips, they struggle with longer content typical in real-world applications like streaming services.

**Key Innovation**: The paper introduces a novel memory system that dramatically improves efficiency by condensing video content into two compact forms:
1. Semantic Memory: Converting video content into concise text descriptions using BLIP-2, a vision-language model
2. Region-Based Memory: Summarizing important visual elements through ROI (Region of Interest) tokens

**Technical Architecture**:
- Built upon MViTv2 (Multiscale Vision Transformer)
- Uses a FIFO (First In, First Out) system to manage memory
- Projects each caption into a single token, significantly reducing dimensionality
- Maintains just 16 memory tokens per video clip
- Implements a captioner-free dynamic strategy for improved inference efficiency

**Performance Advantages**:

*State-of-the-art results on multiple benchmarks*:
- AVA v2.2 action detection
- EpicKitchens-100 action classification
- Breakfast long-term activity detection

*Efficiency Improvements*:

- Uses up to 5x fewer memory tokens compared to baseline models with dense visual memory
- Lower GFLOPs that scale more efficiently with text and region tokens

**Strengths:**

**Originality**:
- The paper introduces the concept of using text as a compression mechanism for video content, which, as far as I know remained unexplored thus far.
- Also introduced is a hybrid memory architecture combining semantic text tokens and ROI visual tokens.

**Rigorous Evaluation**: Results on multiple datasets as well as clear ablations validating design choices.

**Weaknesses:**

**Typos and Language Issues**
- There are typos in Figure 3, where “captioenr” should be “captioner,” and on lines 77 and 78 ("illustarted" should be "illustrated").
- Review the use of adverbs and certain descriptors; for instance, “Nevertheless” in line 539 is redundant as it follows a sentence already commending the model. Similarly, "notably" and "more importantly" are used excessively or inappropriately (e.g., lines 83, 245, 295). Words like "fuels," "outweighs," and "overwhelms" are not appropriate in the context they are used in the paper. Consider revising for clearer emphasis.

**Claims vs. Evidence**
- The paper claims that current methods fail in real-world settings but lacks proof that TinyMem overcomes these challenges in real-world scenarios. Strengthen this by providing benchmarks or examples of such cases.

**Clarity and Structure**
- The introduction is too detailed, detracting from the paper's focus. Consider summarizing and moving background information to a separate section.
- Figure captions (Figures 2 and 3) should summarize each figure’s purpose and insight rather than simply stating what it is.
- The *Methods* section could benefit from a more cohesive structure. Consider how each consecutive subsection follows from the previous one instead of only detailing the concept. (For instance, What is the input of your MULTIMODAL MEMORY EMBEDDING? Where does its output goes next? ...)
- The *Experiments* section is hard to follow. AVA results appear in both Section 4.1 and 4.3, making the results difficult to trace, please consolidate AVA results into a single section.
- Also, the authors are presenting ablations in both sections 4.1 and 4.3 making it hard to know which section is presenting what. Consider a clear separation between "Results" and "Ablations".
- Table 5’s “Vid/s” is ambiguous; specify its meaning for clarity.

**Comparative Analysis**
- Comparison on Epic-Kitchens-100 is only against two methods. Expanding this to include a broader range of baselines would make the result more convincing.

**Technical Details**
- TinyMem is described as a “lightweight alternative,” but it relies on an off-the-shelf captioner and has more trainable parameters than others. Provide more details on FLOPs and throughput relative to baselines for a clearer comparison of its efficiency.
- Novelty appears limited as the main innovation is the use of an off-the-shelf captioner (BLIP) for marginal performance gains. Each frame requires BLIP-2 captioning, creating potential bottleneck. Please add captioning time analysis and address the novelty issue.
- It's not clear the idea behind captioning using a VLM model (BLIP) and then reverse the captioning by using yet another VLM model (CLIP). If BLIP-2 is already pre-trained on text-image pairs, using intermediate representations could improve efficiency rather than captioning each frame and then re-encoding. Could the authors explain the rationale behind that?
- Unclear how the model will scale with other captioners. Consider experimenting with a video-captioning model or exploring alternatives that could enhance the model’s scalability and efficiency.
- In lines 205-207, there are these sentences: *We employ RoI features as compact region representation and feed them into the classification head to gain the final prediction. In consequence, we obtain resulting N_region region memory tokens.* which means that the final prediction is the same as the region memory tokens. Could you clarify this?
- The choice of using only FIFO memory structure should be justified or explained.

**Questions:**

See weaknesses

---

### Official Review · Reviewer_LV1X · 2024-11-03

**Soundness:** 3
**Presentation:** 3
**Contribution:** 3
**Rating:** 6
**Confidence:** 4

**Summary:**

This paper introduces TinyMem, a novel approach to long-form video action detection that addresses the limitations of existing video transformer models that rely on dense visual memory embedding. Rather than using hundreds of memory tokens to capture long-range dependencies, TinyMem employs a more efficient multimodal memory system that combines condensed visual region embedding with abstract text semantics derived from video content. By leveraging vision-language models to generate framewise captions and utilizing ROI features or global tokens for region embedding, TinyMem achieves state-of-the-art performance while using significantly fewer memory tokens than previous approaches. Results are reported on AVA-v2.2, Epic-Kitchens-100 and Breakfast datasets.

**Strengths:**

Strengths:

1. The idea is simple but innovative and well motivated.
2. The paper is well presented and easy to follow.
3. The ablations are detailed and informative.
4. The method achieves strong performance on multiple benchmarks.

**Weaknesses:**

I am concerned regarding the sensitivity of the method on the type of text captioning model/language model being used. As the paper mentions, *"Text embedding overwhelms other formats of embedding on AVA by a large margin"*. I wonder how this varies with different pretrained language models and vision-language models. Additionally, it can be seen that improvements on other benchmarks such as Epic-Kitchens-100 and Breakfast are much less compared to those on AVA. Is that because on Epic-Kitchens-100 and Breakfast the text embedding is not as useful as on AVA? If that is the case, then it would imply the main performance improvements, especially in AVA, is dependent on the quality of the text embedding. Which in turn means that performance is dependent more on the type of pretrained model used and not the actual method being proposed in the paper.

**Questions:**

Please consider the weaknesses section and consider the points regarding impact of the choice of pretrained model. I will be revising my rating after discussing further with the other reviewers.

---

### Note · Authors · 2024-11-21

I have read and agree with the venue's withdrawal policy on behalf of myself and my co-authors.